# Modeling Receptor Motility along Advecting Lipid Membranes

**DOI:** 10.3390/membranes12070652

**Published:** 2022-06-25

**Authors:** Matteo Arricca, Alberto Salvadori, Claudia Bonanno, Mattia Serpelloni

**Affiliations:** 1The Mechanobiology Research Center, University of Brescia (UNIBS), 25123 Brescia, Italy; m.arricca@unibs.it (M.A.); c.bonanno@unibs.it (C.B.); mattia.serpelloni@unibs.it (M.S.); 2Department of Mechanical and Industrial Engineering, Università degli Studi di Brescia, via Branze 38, 25123 Brescia, Italy; 3Department of Civil, Environmental, Architectural Engineering and Mathematics, Università degli Studi di Brescia, via Branze 43, 25123 Brescia, Italy

**Keywords:** mechanobiology, receptor motility, advecting membranes, multiphyisics methodologies

## Abstract

This work aims to overview multiphysics mechanobiological computational models for receptor dynamics along advecting cell membranes. Continuum and statistical models of receptor motility are the two main modeling methodologies identified in reviewing the state of the art. Within the former modeling class, a further subdivision based on different biological purposes and processes of proteins’ motion is recognized; cell adhesion, cell contractility, endocytosis, and receptor relocations on advecting membranes are the most relevant biological processes identified in which receptor motility is pivotal. Numerical and/or experimental methods and approaches are highlighted in the exposure of the reviewed works provided by the literature, pertinent to the topic of the present manuscript. With a main focus on the continuum models of receptor motility, we discuss appropriate multiphyisics laws to model the mass flux of receptor proteins in the reproduction of receptor relocation and recruitment along cell membranes to describe receptor–ligand chemical interactions, and the cell’s structural response. The mass flux of receptor modeling is further supported by a discussion on the methodology utilized to evaluate the protein diffusion coefficient developed over the years.

## 1. Introduction

The present manuscript focuses on the multiphysics modeling of protein motility along advecting animal cell membranes, overviewing the state of the art and proposing suitable physical laws to couple receptor relocation on membranes with cellular mechanical deformation. From a conceptual point of view, physical theories and mathematical tools allow us to relate the mechanical principles with the behavior of living matter: thermo-mechanics of continua [1,2] is the ideal framework to model nature’s laws. Due to its intrinsic interdisciplinarity, a multi-physics approach to biological phenomena may have the potential to highlight key and limiting factors, providing innovative pathways for analysis and interpretation.

Receptor motility along cell membranes is involved in several biological processes, such as cell, bacteria, and virus adhesion and motility, as well as endocytosis and exocytosis, to cite a few [3,4,5,6,7,8]. The cell membrane plays a crucial role in cellular protection, in the control and transport of nutrients [9], and in regulating the interchange of different substances in the cell [10]. Its structure facilitates directional or Brownian diffusion of receptors, internalization, and segregation. Acting as a barrier between the extra-cellular and intra-cellular environments, the cell membrane controls the flux of matter across and along its surface [11]. Being constituted of two sheets of phospholipid (amphoteric) molecules, cell membranes in aqueous environments acquire the conformation of a phospholipid bilayer, with the hydrophobic end inside the bilayer and the hydrophilic outside. Such a conformation, including the various embedded proteins (receptors, ion channels, transporters, and other proteins), constitutes the so-called *fluid-mosaic* model [12]. Membrane fluidity represents one of the most critical membrane properties [9], and it is still the object of several studies [9,10,11,13,14,15,16,17]. Among numerous molecules relocating along the plasma membrane, we focus here on the motility of receptor proteins. For the sake of generality, and due to the aim of this manuscript, we do not distinguish among the different types of receptors in terms of structure and second-messenger systems.

The paper is organized as follows. Continuum models of receptor motility along advecting membranes are illustrated in Section 2; in it, we will collect and discuss publications on receptor–ligand-mediated cell adhesion, cell contractility, protein relocation on advecting membranes, receptor-mediated endocytosis, and a few other subjects. Statistical models of receptor dynamics are shortly recapitulated in Section 3. In view of the huge number of publications in this realm, we cannot aim at being exhaustive. In the following Section 4, we discuss some specific thermodynamic prescriptions for the mass flux of receptors and the kinetics of receptor– ligand interactions. The summary in Section 5 completes the manuscript.

## 2. Continuum Models of Receptor Motility

### 2.1. Receptor–Ligand-Mediated Cell Adhesion

Receptor–ligand-mediated cell adhesion is one of the most common and widely studied mechano-biological processes. In it, receptor motility plays a significant role. To the best of our knowledge, the pioneering studies of Bell [18] and co-workers [19] paved the way in developing multi-physics models in mechanobiology. The receptor density in receptor–ligand-mediated cell–cell adhesion was analyzed in a thermodynamic framework, investigating the competition between attractive receptor–ligand and repulsive electrostatic forces. The former was proven to be of greater influence [18]. An increment of receptor concentration in the adhesion zone was further proposed as a transduction mechanism for triggering different cellular responses. Similarly, the redistribution of receptors is viewed as a signal for cell polarization. Phase transitions occur in cell adhesion, and the stabilization of the cell–cell contact is achieved via cooperative rearrangements of the internal components of the cells [19] (Bell and co-workers [19] used *internal variables* literally, instead of *internal components*. As for other terminology such as *plasticity*, this example displays how very similar words have completely different meanings in mechanics and biology. Such an outcome of the cultural and historical evolution of the disciplines is a further challenge in mechanobiology). Goldstein et al. [20] published a theoretical study of the interaction of low-density lipoprotein (LDL) receptors with coated pits (specialized cell surface structures in which receptors aggregate). They evaluated the diffusion limits for the forward rate constant of the receptor–ligand chemical interaction on a human fibroblast, as well as the average time that LDL receptors spend on the cell surface before being trapped in a coated pit. The obtained results, in agreement with the experiments, led them to conclude that if LDL receptors are inserted at a random position in the cell membrane, their motion is driven by pure diffusion before being trapped in coated pits. A further study [21] found that the way in which coated pits return to the surface does not affect the average time that receptors spend on the membrane, the forward rate constant or the fraction of receptors aggregated in coated pits at high values of the diffusion coefficient, whereas the effect is substantial for “immobile” receptors.

To mimic cell–tissue interaction, the kinetics of cell adhesion due to the bonding between $\alpha_{\mathrm{II}b}\;\beta_{3}$ integrins and ligands, gravitation, and Helfrich repulsion were studied in [22] for a single giant vesicle on a solid substrate. The analysis of the growth of the adhesion front revealed the prominent role of the receptor–ligand pairs: at high concentrations, the kinetics of ligand–receptor formation drives the propagation of the front of (tight) adhesion at constant velocity, whereas small ligand densities entail a diffusion-limited growth with a square root dependence on the time. The role of receptor motility in the process of adhesive contact was analyzed in the transient growth of the adhesion zone by Freund and Lin [23]. They assumed the flow of receptors to be proportional to the local gradient in chemical potential and formulated a continuum model of the adhesion of an initially uniformly curved elastic plate to a flat substrate. For very large plates, they solved the problem in closed form, whereas the necessity of numerical methods emerged for those of a limited size. Using the same model, Shenoy and Freund [24] investigated the expansion of a circular adhesion zone when binder (ligand) density is insufficient to overcome the repulsive barrier that resists cell adhesion. They explained the cross-over effect observed in [22] when the densities of ligands and receptors are equal. Indeed, the growth of the adhesion front radius with a time square root dependence observed experimentally in [22] was recovered
(1)$$R\left(t\right)=2\;a\;\;\sqrt{{D\;|\;}_{L}\;t}\;,$$
with *t* meaning time, ${D\;|\;}_{L}$ ligand diffusivity and *a* coefficient depending on the ratio between the ligand and receptor concentrations, $c_{L}$ and $c_{R}$; particularly, *a* assumes finite values for $c_{L}/c_{R}<1$, whereas the square-root growth regime breaks off for $c_{L}/c_{R}≃1$.

Liu et al. [25] extended the former framework [23,24] and introduced a so-called traction–separation relation to model cell–substrate interaction. They provided an additional contribution to the flux of receptors, otherwise governed by the classic Fick’s law, proportional to the traction component tangent to the membrane surface, to account for the role of non-specific force as driving force for the recruitment of receptors towards the adhesion front. Numerical simulations via finite-element methods shown that the advancing adhesion front might be stable or unstable if exposed to small perturbations, as a function of the membrane shear modulus, the adhesive tractions, and the receptor density. Instability occurs at high adhesive tractions, soft membranes or high ligand–receptor concentration ratios. The traction–separation model [25] was extended in [26], performing simulations of biotin receptor-streptavidin ligand binding-mediated attachment–detachment of a red blood cell to a substrate. A surface diffusion model described receptor motility. The governing equations were implemented in a finite element scheme, providing results in agreement with experimental data.

Golestaneh and Nadler [27] introduced a spontaneous area dilation to account for the influence of receptors on cell deformation and adhesion. Similar to the adhesion–traction model [25,26], a non-linear receptor–ligand binding force replicated the charge-induced dipole interactions, while Fick’s law governed the diffusion of the receptors on the membrane. This study examined the nature of the coupling between electrostatic adhesive forces and the deformation of particle [28] via a non-linear continuum model. A strong coupling was found for small and moderate membrane deformations.

### 2.2. Cell Contractility

Deshpande and co-workers proposed a bio-mechanical model, widely used later on, to couple cell contractility with focal adhesions (FAs) [29]. The mechano-sensitive properties of FAs were modeled in a continuum framework, wherein the cytoskeletal contractile forces generated by stress fibers (SFs) drive and stabilize the assembly of the FA complexes. The model accounts for the diffusion of low-affinity integrins along the cell membrane and predicts different levels of concentration of FAs. Simulations replicated high concentration of FAs around the periphery of the cell, the increment of FAs at decreasing cell sizes, and the decrease in intensity of FAs if cell contractility is curtailed. Stemming from this framework, a signaling model was devised based on the generation of ${\mathrm{IP}}_{3}$ molecules during the FA growth [30], predicting the range of ${\mathrm{IP}}_{3}$ diffusivities at which the SF activation signal is spatially uniform. The model [29] was also employed for investigating the role of actin cytoskeleton in compression and cell adhesion [31,32], and to account for the feedback between intracellular signaling, FA formation, and SF contractility in the osteoblast response on a grooved substrate [33]. Simulations revealed the presence of stretched SFs dominant bundles during compression for polarized and axisymmetric spread cells. Round cells were predicted to have fewer SFs and a lower compressive strength. Highly contractile cells were revealed to provide greater resistance to compression by means of dominant circumferential SFs [31]. Supported by experimental observations, the substrate-dependent response of contractile cells with no predefined SF or FA arrangement was predicted. SF contractility was found to affect the substrate-dependent response of cells, including changes in nuclear stress and cell tractions. An increment in SF and FA formation was numerically predicted for stiffer substrates [32]. In [33], it was shown that the cell orientation is governed by the diffusion of signaling proteins activated at FA sites on the ridges. The responsiveness of osteoblasts to the topography of substrates was rationalized by the model. Broadening [29,33], a non-local finite element setting, was implemented in [34] to study the competition between cytoskeletal and passive elastic-free energies as a driving mechanism in cell spreading. As experimentally observed, a high concentration of aligned SFs along free edges corresponds to a state with low free energy. McMeeking and Deshpande [35], while summarizing previous models [36,37], presented a bio-chemo-mechanical model implemented in a finite element code for simulating in vitro cell behavior. They targeted contractility, adhesion, signaling, cytoskeleton formation and remodeling.

A coupled formulation of chemo-diffusive integrins with the cytoskeleton, underlying cell contraction and spreading, was proposed in [38]. In agreement with experimental observations, numerical simulations suggested that substrate stiffness and chemistry strongly affect cellular contraction and spreading. The relevant role of mechanics in contraction, adhesion, and spreading of adherent cells was highlighted.

### 2.3. Protein Relocation on Advecting Membranes

Mikucki and Zhou [39], using an energetic variational principle on advecting membranes, derived a curvature-driven transport equation, relating molecule concentrations to a gradient flow governed by a drift-diffusion equation. They predicted the molecular localization on static membrane surfaces at locations with preferred mean curvatures, and that the molecular localization is in turn driven by the generation of preferred mean curvature.

Carotenuto et al. [40] developed a multi-physics approach to investigate how ligand–receptor interactions along the cell membrane trigger raft formation. Diffusion and kinetics of binding and unbinding were studied. Understanding how transporters and active receptors trigger raft formation and clustering is of paramount relevance in membrane-mediated phenomena such as COVID-19 virus–cell interaction.

A discrete model of chemotaxis, which takes into account possible alterations in cellular motility, was presented in [41]. A derivation of the Patlak–Keller–Segel (PKS) model as a continuum limit from the discrete model was shown. Comparisons between numerical simulations of the discrete model and numerical solutions of the PKS model were performed, showing an excellent agreement between the two models.

The authors of this review studied the relocation of transmembrane receptors along advecting cell membranes, like for Vascular Endothelial Growth Factor Receptors 2 (VEGFR2) and $\alpha_{v}\beta_{3}$ integrins, by designing chemo-transport-mechanical multi-physics formulations [42,43,44,45] to describe how the mechanical behavior of an endothelial cell (EC) affects receptor dynamics during the early phases of tumor angiogenesis. VEGFR2 dynamics on cell membranes was studied in [42] for EC adhesion onto a rigid substrate coated with specific immobilized ligands, on the basis of the established role as activator of the angiogenic process of the chemical interactions between soluble non-canonical ligands, as gremlins [46,47], released by cancer cells. Although strongly simplified assumptions on cell mechanics were made, VEGFR2 dynamics was well captured and validated against co-designed experimental investigations [48]. The emergence of three different phases of VEGFR2 relocation and complex generation was unveiled and related to distinct mechanisms, including: (i) the initial cell–substrate contact interaction and the VEGFR2–gremlin high chemical reaction rate, (ii) the mechanical deformation, (iii) the VEGFR2 relocation on EC membrane due to diffusion. The mathematical description of the model was detailed in a companion paper [49]. The model, framed in the mechanics and thermodynamics of continua, follows a general description proposed in [50], and takes advantage of successful descriptions of physically similar systems [51,52,53]. The model has been broadened [43,44] to account for the interplay between VEGFR2 and VEGF-A or thegremlin, $\alpha_{v}\beta_{3}$ integrin and the glycoprotein fibronectin embedded in extracellular matrix, and the experimentally revealed interaction between $\alpha_{v}\beta_{3}$ integrin and the VEGFR2–gremlin complex [54]. The induced receptor polarization was identified in cell protrusions and in the basal aspect of ECs. Relocation and reaction of $\alpha_{v}\beta_{3}$ receptors along cell membranes were also included in a general framework for cell spreading, motility, and receptor dynamics [44,45]. The mechanics of the cell was accounted for in the field of finite strain theory in continuum mechanics and in a consistent (continuum) thermodynamic setting, together with the modeling of relocation and reaction of actin proteins to form biopolymer structures.

### 2.4. Receptor Mediated Endocytosis

Based on [23,24], Gao, Shi and Freund [55] presented a receptor-mediated endocytosis study, considering the role of mobile receptors in wrapping the cell membrane around a cylindrical or spherical particle coated with immobile and uniformly distributed ligands. They showed the existence of a minimum value of both particle radius and receptor density below which wrapping cannot take place. An estimation of the size of the smallest and the largest particle that can be successfully wrapped was given.

A similar study was performed by Decuzzi and Ferrari [56]. They considered both elliptical and cylindrical particles, showing how the internalization is affected by size and aspect ratio.

The same model proposed in [55] allowed us to develop a framework for modeling uptake and release of nanoparticles in human and animal cells. In that paper, the mechanics of cell–nanomaterial interactions was investigated, showing how nanoparticles enter cells by receptor-mediated endocytosis. Coarse-grained molecular dynamics was implemented to perform simulations of nanoparticles interacting with cell membranes [57].

Further works on receptor-driven endocytosis were presented by Wiegold, Klinge, Gilbert, and Holzapfel [58,59]. They considered viruses as a substrate with fixed receptors, whereas receptors of the host cell could relocate on its membrane. Numerical simulations performed via finite difference methods showed a rapid variation in receptor density at the early stage, while approaching a steady state as the time progresses.

### 2.5. Protein Motility Miscellanea

Lee et al. [60] proposed a finite-difference mathematical model to describe charged receptor transport on the cell membrane, showing the importance of cell shape in receptor diffusion and in the response to an extracellular sinusoidal electric field. They illustrated how the distribution of receptors may alter transmembrane potential and highlighted the prominence of cell shape (i.e., of the mechanics that rules its evolution) in governing interactions between alternating current electric fields and receptors.

Mac Gabhann and Popel [61] modeled the effect of placental growth factor (PlGF) on the response of VEGF ligands in pathological angiogenesis. A set of coupled reaction–diffusion equations described secretion, transport, binding, and internalization of ligands. The presence of PIGF was established to determine a change in the formation of endothelial surface growth factor–VEGFR1 complexes, and a less significant increment in the number of VEGFR2 complexes. Similar equations were used in [62] to study the binding kinetics and signaling pathways of basic fibroblast growth factor (FGF-2) through a reaction–diffusion model of in vitro FGF-2 transport and receptor–ligand binding. Based on experimental results that included degradation of the internalized cell surface species, formation of double triads, and dimerization of FGF-2 ligands, the role of the low-affinity heparan sulfate proteoglycans (HSPGs), the identity of the minimal signaling complex leading to FGF-2 activity, and the importance of FGF-2 dimerization were pointed out.

Rattanakul, Crooke et al. [63] modeled the signal transduction pathways involving G-proteins by including reaction–diffusion equations of various reactants both inside and on the extra-cellular surface membrane. They investigated the dynamic and steady-state properties of the model via weakly nonlinear stability analysis, showing the robust formation of Turing-type patterns under different system parameters, and discussing theoretical predictions against reported experimental evidence.

As an extension of [64], Earnshaw and Bresslof used reaction–diffusion equations to describe the trafficking of α-amino-3-hydroxy-5-methyl-4-isoxazolepropionic acid receptors (AMPA-Rs) and to evaluate how lateral diffusion contributes to the strength of a synapse [65]. They calculated the distribution of synaptic receptor numbers across the population of spines, determining the effect of lateral diffusion on the strength of a synapse.

Daniels [66] deduced a mathematical expression, in the perturbative deformation regime, to describe the diffusion-limited reaction rate. The coupling between the deformation of a curved membrane and the chemical activities along it was accounted for. The reduction of 20% of the receptor–ligand reaction rate due to the locally induced membrane curvature was theoretically derived.

## 3. Statistical Models of Receptor Motility

Kusumi et al. [67] studied the relocation of E-cadherin and transferrin receptors along mouse keratinocyte cell membranes. A compartmentalization of the cell membrane in small domains, wherein receptors are confined, was suggested as a consequence of the detection of four types of receptor motion (stationary mode, simple Brownian diffusion, and directed and confined diffusion). This conjecture arose from the development of a mean-square displacement (MSD)-based method and the experimental comparison between single-particle tracking (SPT) and fluorescence photobleaching recovery (FPR).

In investigating the non-Brownian diffusion of molecules on membranes by the STP method, Monte Carlo simulations on particles undergoing short-term confined and long-term hop diffusion within a compartment were performed. This simulation strategy detects and characterizes the anomalous diffusion by systematically varying the frame time and rate [68].

By means of a coarse-grained triangular element model, Atilgan and Sun [69] developed a Monte-Carlo methodology, examining the changes in free energy during membrane shape transitions. They showed how a critical value of the concentration of proteins may bring to the formation of small vesicles, therefore influencing the topology of the plasma membrane.

A bimolecular fluorescence complementation (BiFC)-based approach combined with fluorescence correlation spectroscopy (FCS) was used to monitor the diffusion of G-protein-coupled receptor oligomers in the plasma membrane [70]. The approach was used for the first time to measure the membrane diffusional characteristics of adenosine $A_{1}$ and $A_{2A}$ receptor homo- and heterodimers in Chinese hamster ovary cells, demonstrating the differences in diffusivity between adenosine receptor homo- and heterodimers.

Paszek et al. [71] developed a chemo-mechanical model in which integrin diffusion, changes in integrin activation status, and integrin–ligand interactions were simulated via kinetic Monte Carlo (KMC) algorithms. The results show the mediator role of glycocalyx in integrin–ligand interactions, which was found to be sufficient to drive integrin clustering even in the absence of cytoskeletal crosslinking or homotypic integrin–integrin interactions.

Receptor dynamics was also accounted by Duke and Graham [72] in reviewing statistical mechanical models for receptors clustering. They accounted for cluster generation and discussed the equilibrium thermodynamics of receptors, ligands, and cytosolic adaptor proteins. The role of adaptor proteins in permitting cells to exert control on cluster formation and to target clustering at specific locations on the cell surface was highlighted.

A nano-meter scale mathematical model that couples membrane bending and long surface molecules (LSMs) compression was presented in [73] to reproduce the lateral mobility of LSMs by drift–diffusion equations. Size-based segregation of LSMs from a receptor–ligand complex was proposed as the mechanism of receptor triggering. Supra-diffusive segregation of LSMs from a single receptor–ligand complex was found.

The reduced mobility of receptors after aggregation processes on the membrane was modeled via both standard and density-dependent diffusion equations in [74]. Critical values of the mobility were compared with numerical simulations, showing that the formation of the aggregate is quite influenced by density-dependent diffusion.

Martini et al. [75] studied the kinetics of a membrane-integrated protein that locates at specific binding sites on the genome, and also acts as a transcriptional activator. Mathematical analysis and KMC simulations of lattice models were combined with fluorescence-microscopy experiments. CadC (the pH receptor of the acid stress response Cad system in E. coli) diffusion along the membrane and conformational fluctuations of the genomic DNA were accounted for. They found that diffusion and captured mechanisms are potentially sufficient for bacterial membrane proteins to establish functional contacts with cytoplasmic targets.

## 4. Discussion

### 4.1. Modeling the Mass Flux of Receptors

Most continuum models described in Section 2 account for the protein transport. The mass flux of receptors, $h_{R}$, is usually described via Fick’s law,
(2)$$h_{R}=-{D\;|\;}_{R}\;\nabla_{{\;}_{S}}\left[\;c_{R}\;\right]\;,$$
with diffusivity ${D\;|\;}_{R}$ and surface gradient operator $\nabla_{{\;}_{S}}\left[\;\;\;\right]$.

Other authors [25,26] reformulated Equation (Equation 2) for $h_{R}$, assuming that receptors are attracted by ligands by means of a traction force function of the receptor-l-igand distance—within a certain cutoff distance—and concentrations. In this case, in addition to Fick’s law concentration gradient, a further flux term
(3)$$h_{R}^{T}=u_{R}\;t\cos\beta$$
shall be accounted for, where $u_{R}$ is the receptor mobility under ligand attractive forces and tcosβ represents the tangential component of the traction—see Figure 1.

The influence of non-specific traction forces exerted by ligands on receptor motility and cell adhesion has to be related to the cell size and/or the stage of adhesion considered. Studies performed on different time scales and cell sizes, for the spreading of a mouse embryonic fibroblasts on a matrix-coated surface [76], or for a bovine aortic EC on polyacrylamide gels [77], confer an influence of such forces mostly related to the early stages of adhesion. The adhesion–traction model predicts the isotropic early stage of cell adhesion [25] well, essentially independent to cytoskeleton remodeling and strongly dependent on high ligand densities. This was made even more clear in [26], where a micropipette-manipulated red blood cell attachment–detachment model showed, for an analysis of ≈50 ms, a level-off of the adhesion–traction forces after approximately a third of the adhesion-spreading time, and a consequent need for receptor diffusion from remote areas of the cell to fuel the spreading. Moreover, at lower densities of ligands, the spreading becomes strongly dependent on cytoskeletal structures, and cells tend to spread anisotropically by randomly extending pseudopodia [77]. Furthermore, studies on charged flexible particles that adhere to an oppositely charged rigid substrate due to electrostatic attraction forces [28] established that surface forces drive small particle adhesion. The underformed cell radius was considered in the micron/sub-micron range 1 μm [77], or even smaller, 12.5 nm [27]. It follows that, in dealing with cells of radius 10 μm or higher (such as ECs), receptor motility and cell adhesion can be considered unaffected by short-range surface tractions, since their energetic contribution appears to be insufficient in cell spreading without accounting for pseudopodia migration mechanisms.

### 4.2. Evaluation of the Protein Diffusion Coefficient

The diffusive motion of a particle α is predicted by the well-known Einstein– Smoluchowski relation,
(4)$${D\;|\;}_{\alpha }={u\;|\;}_{\alpha }\;k_{B}\;T$$
with $k_{B}$ as the Boltzmann constant, ${u\;|\;}_{\alpha }$ mobility coefficient and *T* temperature. To the best of our knowledge, the first model to evaluate the diffusivity was proposed by Saffman and Delbrück (SD) [78], describing the diffusion of a particle due to Brownian motion in biological membranes, and demonstrating a weak logarithmic dependence of the lateral diffusivity on the particle radius $r_{\alpha }$,
(5)$${D\;|\;}_{\alpha }\propto \ln\left[\;r_{\alpha }\;\right]\;.$$

Several experiments have been devoted to investigating the parameters that influence the diffusivity of proteins on the membrane, such as the membrane’s thermal fluctuations [79], the bending rigidity and surface tension [80], the change in the membrane shape [81], and the hydrophobic mismatch between protein length and membrane thickness [82,83]. The Saffmann theory [84] of membrane hydrodynamics was extended to investigate the correlated Brownian motion of protein pairs [85,86]. The influence of protein concentration on the motion was accounted for in deriving expressions for the diffusion coefficients as a function of concentrations for small protein size [85]; the effect of the immobile inclusions on the membrane was studied in [86].

The Saffman–Delbrück theory [78] was questioned when investigating the dependence of ${D\;|\;}_{\alpha }$ on the protein radius [87], because a non-hydrodynamic primary source of protein drag was found after experimental observations. A numerical framework to predict the diffusivity of arbitrarily shaped objects embedded in lipid bilayer membranes was proposed [88], and the influence of finite-size effects in molecular simulations was investigated [89]. An underestimation of diffusion constants due to the sizes of the simulation was predicted via coarse-grained Martini and all-atom CHARMM36 (C36) force fields [90]. By measuring the lateral mobility of transmembrane peptides via fluorescence correlation spectroscopy (FCS), the SD theory was confirmed for low protein-to-lipid ratios, whereas a linear dependence between the diffusion coefficient and the protein radius was found for higher protein-to-lipid ratios [91], further accounting for the influence of the peptide structure by comparison between experimental data and coarse-grained molecular dynamics (MD) simulations [92]. How peptide–membrane lipid interactions generate mechanisms that drive membrane deformations and lead to the curvatures necessary in membrane remodeling processes was examined in [93]. The surface chemistry adaptability of peptides via side chain rearrangements in response to the environment, the amplification of their activity by means of hydrophobicity and cationic charge, and the pivotal role of their shape-changing properties in interacting with membranes was demonstrated [94]. The collective diffusion coefficient in deformable bilayer membranes hosting transmembrane proteins that diffuse collectively was studied, revealing the resistance exerted by the presence of proteins on monolayer sliding [95]. Via fluorescence correlation spectroscopy, diffusion coefficients of transmembrane proteins in different types of membrane were estimated in [96], and a linear decreasing trend of membrane-bound protein diffusivities with the increase in membrane coverage proteins was proved in [97].

The discrepancies in the evaluation of the diffusion coefficient with that predicted by the SD model results were analyzed in [98]. However, the mobility of a rigid spherical particle of radius $r_{\alpha }$ in a 3D solvent with viscosity ν has an inverse dependence on the particle radius, as found by the SD theory [78] and expressed by the well-known Stokes–Einstein relation,
(6)$${u\;|\;}_{\alpha }=6\pi \;\nu \;r_{\alpha }\;,$$

The mobility of the same particle when embedded in a 2D fluid membrane is further elaborated. Complexities arise from the coupling between the 2D fluid and the surrounding 3D solvent, with the constraint of no slip at the interfaces. Such hydrodynamic coupling introduces an inherent length scale into membrane hydrodynamics, $\zeta =\nu_{{}_{S}}/\nu$, where $\nu_{{}_{S}}$ is the 2D membrane viscosity. It was therefore proposed to evaluate the mobility of a diffusive particle on a membrane according to [98]: (7)$${u\;|\;}_{\alpha }=\frac{1}{4\pi \;\nu_{{}_{S}}}\;f\left(\zeta /r_{\alpha }\right)\;\mathrm{with}\;f\left(\zeta /r_{\alpha }\right)=\left\{\begin{array}{cc} \pi \zeta /4\;r_{\alpha } & \mathrm{for}\zeta \ll r_{\alpha }\\ \ln\left[\;\zeta /r_{\alpha }\;\right]-\gamma_{E} & \mathrm{for}\zeta \gg r_{\alpha } \end{array}\right.$$
with $\gamma_{E}$ is the Euler–Mascheroni constant.

In conclusion, the results show that the weak logarithmic dependence (see Equation (Equation 5)) of the diffusivity on $r_{\alpha }$ in fluid membranes holds particles much smaller than ζ. For proteins, the applicable limit is $\zeta /r_{\alpha }\gg 1$, suggesting that all membrane-bound proteins, and even the constituent lipids of the membrane, should have approximately the same diffusion constant [98].

### 4.3. Modeling Receptor–Ligand Kinetics

The chemical reaction
(8)$$R+L\overset{k_{\mathrm{on}}}{\underset{k_{\mathrm{off}}}{⇄}}C$$
portrays the conversion of freely diffusive receptors on the membrane to trapped receptors, and vice versa. $k_{\mathrm{on}}$ and $k_{\mathrm{off}}$ are the forward and reverse rate factors for the formation, or dissociation, of the complex C from free receptors and ligands, R and L.

The authors of this review modeled the reaction rate *w* of the chemical reaction (Equation 8) through the following law of mass action [99]
(9a)$$w=k_{\mathrm{on}}\;\frac{\theta_{R}}{1-\theta_{R}}\;\frac{\theta_{L}}{1-\theta_{L}}-k_{\mathrm{off}}\;\frac{\theta_{C}}{1-\theta_{C}}\;,$$
where $\theta_{\alpha }$ is defined as $\theta_{\alpha }=c_{\alpha }/c_{\alpha }^{\;max}$, with $c_{\alpha }$ meaning concentration (molecules per unit area) and $c_{\alpha }^{\;max}$ concentration saturation limit, for α=R,L,C.

Another common way to write Equation ([Disp-formula FD9a-membranes-12-00652]), as in [18,26], for instance, is
(9b)$$w=k_{\mathrm{on}}\;\left(c_{R}-c_{C}\right)\;\left(c_{L}-c_{C}\right)-k_{\mathrm{off}}\;c_{C}\;.$$

Equation ([Disp-formula FD9b-membranes-12-00652]) appears in [18,26] written in terms of densities (number of species per unit area) instead of concentrations. The relation which links the density of species, $\rho_{\alpha }$, to the concentration is
$$\rho_{\alpha }=\kappa_{\alpha }\;c_{\alpha }\;,$$
with $\kappa_{\alpha }$ molecular density.

Considering that external forces may cause the unfolding or disruption of receptor–ligand complexes [18,19,100,101], it was proposed in [26] to augment the law of mass action ([Disp-formula FD9b-membranes-12-00652]) by two exponential terms. The first exponent multiplies the forward reaction term of the equation and depends on the receptor–ligand separation distance (within the cutoff limit). The latter, multiplying the backward reaction term, is assumed to be dependent on the ligand–receptor traction force. Equation (9) arises considering only short-range receptor–ligand interactions, which lead to strong adhesion, much more than non-specific forces [18,25].

The chemical reaction (Equation 8) descends from a more general relation that describes the generation of a protein complex via a two-step mechanism in which the formation of an encounter complex R|L precedes either the generation of the final complex C or the recovery of free proteins [102,103]: (10)$$R+L\overset{{k_{\mathrm{on}}}^{\prime }}{\underset{{k_{\mathrm{off}}}^{\prime }}{⇄}}R|L\;\overset{k_{\mathrm{on}}}{\underset{k_{\mathrm{off}}}{⇄}}C\;,$$
where the coefficients ${k_{\mathrm{on}}}^{\prime }$ and ${k_{\mathrm{off}}}^{\prime }$ represent the rate of formation and dissolution of the encounter complex.

In the formation of R|L, proteins change their orientations, leading either to evolution into C, when R and L proteins match to each other, or to dissociation into free proteins. Geometry, in terms of inter- protein distances and rotation angles with respect to the orientation of C, is relevant in achieving the final step of association. In a certain range of distances and angles, an electrostatic steering region determines the directional diffusion mobility of protein(s) instead of Brownian motion [103].

If the concentration of R|L is smaller than the concentration of both the free proteins and the final complexes, it is a good approximation [18] to neglect the variation in time of R|L in Equation (Equation 10), leading to the most commonly used relation (Equation 8). Accordingly, the generation of a receptor–ligand complex can be considered to occur immediately once the receptor–ligand distance is sufficiently small. This allows us to disregard the dependence upon the cutoff distance and the predominant rotational Brownian motion of receptors in the R|L state [78]. Tight cell–substrate adhesion allows us to consider complexes as immobilized to the substrate once generated, leading to the mass action law in the form of Equation ([Disp-formula FD9a-membranes-12-00652]).

Lastly, as introduced in Section 2, receptor motility is commonly associated to cell adhesion and spreading, therefore requiring a description of the cell by means of the laws of mechanics. Cell mechanical response can be assigned either to the bulk of the cell or to the cell membrane. The former choice, supported by [29,30,31,32,34,35,36,37,44,45,104], for instance, implies the assignment of the cell structural response to the cytoskeleton remodeling; alternatively, other authors demand the structural functions to the cell membrane, for instance [105,106,107,108,109]. Despite studies on red blood cells providing a description of the cell membrane deformation at a constant area [110], the influence of curvature on the membrane elastic stiffness is related to cell dimensions. The link among curvature, elastic stiffness and cell dimension was highlighted in [27] in studying a small cell of the radius 12.5 nm.

We are persuaded that the attribution of the structural response to the cell membrane and the importance of binding forces on the mechanical response of cells stated in [27] do not match with cells with larger dimensions, as ECs.

## 5. Summary

In this article, we summarized theoretical approaches and computational methodologies developed since the late 1970s, in modeling protein motion along advecting membranes for different biological processes. It has been our aim to collect some of the most emblematic mathematical and computational methodologies, providing a broad introduction to a scientific topic which is in great development nowadays.

Multi-physics methodologies applied to receptor motility along cell membranes may provide a rationale to the evolution in time of quantities of interest for the protein dimerization processes observed experimentally, identifying limiting factors with significant accuracy. Receptor dynamics and receptor–ligand chemical interactions are coupled with cell mechanics. Mechanobiology provides the description of the evolution of cells, with the potential to predict protein dynamics and cell behavior in biological processes. Co-designing theoretical multi-physics frameworks, numerical simulations, and experimental outcomes may allow us to identify the laws that regulate receptor activation, relocation and recruitment, therefore opening new perspectives to support biological and medical research.

## Figures and Tables

**Figure 1 membranes-12-00652-f001:**
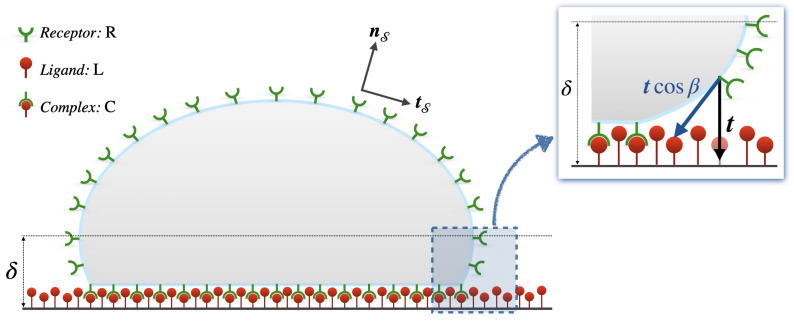
Schematic of an adherent cell onto an enriched ligands substrate, inspired by [25,26]. The image depicts the concept formulated within the adhesion traction model. Vectors $n_{S}$ and $t_{S}$ represent the normal and the tangent vector at a certain location on the cell membrane, respectively. Within the cutoff limit δ, the tangential component tcosβ of the traction exerted by ligands (vertical traction t), attracts the receptors on the cell membrane, generating an additive flux term, $h_{R}^{T}$, appearing in Equation (Equation 3). β is the angle with respect to the vertical defined by t.

## Data Availability

Not applicable.

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
