# Peer review of "Modeling Receptor Motility along Advecting Lipid Membranes"

_membranes, 2022, doi:10.3390/membranes12070652_

Round 1
Reviewer 2 Report
The authors have submitted an interesting article with the title “Modeling Receptor Motility along Advecting Lipid Membranes”. The overall design of the article is good. I recommend publishing this work after a minor revision. Here are my comments:
1. The abstract of the article is too short and the keywords are not matched with the title and the abstract. Maybe after expanding the abstract, the current keywords make more sense.
2. There are several grammatical errors in the text. For example, in line 386 “nawadays”. Revise the manuscript thoroughly
3. Be consistent in using abbreviations: in the context, the authors wrote “Fig. X”, while in the captions wrote “Figure X”.
4. Part 5 “conclusions” should be changed to “summery”.
